# How to Make a Smartphone-Based App for Agricultural Advice Attractive: Insights from a Choice Experiment in Mexico

Janet Molina-Maturano [1], Nele Verhulst [2,*], Juan Tur-Cardona [1], David T. Güerena [2,3], Andrea Gardeazábal-Monsalve [2], Bram Govaerts [2,4], Hans De Steur [1] and Stijn Speelman [1]

1 Department of Agricultural Economics, Faculty of Bioscience Engineering, Ghent University, 259000 Ghent, Belgium; janet.molinamaturano@ugent.be (J.M.-M.); juan.turcardona@ugent.be (J.T.-C.); hans.desteur@ugent.be (H.D.S.); stijn.speelman@ugent.be (S.S.)
2 International Maize and Wheat Improvement Center (CIMMYT), El Batan, Texcoco 56237, Mexico; d.guerena@cgiar.org (D.T.G.); a.gardeazabal@cgiar.org (A.G.-M.); b.govaerts@cgiar.org (B.G.)
3 Big Data Platform for Agriculture, The Alliance of Bioversity and the International Center for Tropical Agriculture (CIAT), The Americas Hub, Cali 763537, Colombia
4 Cornell University, Ithaca, NY 14850, USA
* Correspondence: n.verhulst@cgiar.org

**Abstract:** Mobile phone apps can be a cost-effective way to provide decision support to farmers, and they can support the collection of agricultural data. The digitisation of agricultural systems, and the efforts to close the digital divide and to include smallholders, make data ownership and privacy issues more relevant than ever before. In Central and South American countries, smallholders' preferences regarding data licenses and sharing have largely been ignored, and little attention has been paid to the potential of nonfinancial incentives to increase the uptake of digital solutions and participation by farmers. To investigate incentives for smallholder farmers to potentially use an agricultural advisory app in which they share their data, a Discrete Choice Experiment was designed. Based on a survey of 392 farmers in Mexico, preferences for attributes related to its usage were revealed using a conditional logit (CL) model. To explore heterogeneity, groups and profiles were explored through a latent class (LC) model. The CL model results revealed, for example, farmers' positive preference to receive support at first use and access to training, while negative preference was found for sharing data with private actors. The LC identified three classes which differ in their preference for attributes such as the degree of data sharing. Furthermore, for example, a farmer's connectedness to an innovation hub was found to be one of the significant variables in the class membership function. The main contribution of the study is that it shows the importance of nonfinancial incentives and the influence of data sharing on farmer preferences.

**Keywords:** discrete choice experiment; smallholder farmer; innovation hub; data ownership; preferences; data privacy; extension services; Mexico

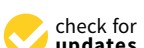



## 1. Introduction

Mobile-phone-enabled agricultural information services are often considered a cost-effective way for providing tailormade services to farmers (e.g., through agricultural advice helplines, SMS and apps, weather forecasts, market information, and mobile finance) while supporting the collection of more comprehensive and accurate statistics (e.g., crop yields and soil information) [1–3]. Thus far, the potential of information and communication technologies (ICT) for agricultural extension is not fully realised. Pushing certain technologies rather than responding to the particular communication challenges of end users is one important cause of that [4]. Therefore, further research for ICT and agricultural extension in the global south should rest on strong user-centredness and problem orientation [4]. Even if the collection and use of farm-level data can improve smallholder farmers' access

to services, there are still concerns around the access, control, and use of data that could lead to smallholder exclusion from benefits [5].

Recently, there has been growing interest in the study of the social implications of digital farming in general and in data ownership and sharing in particular [6]. The lack of trust between the farmers as data contributors and those third parties who collect, aggregate, and share their data was identified as a concern among smart farming and big data participants [7]. It is recognised as one of the main issues that needs to be considered when maximising the impact of ICT interventions in agriculture [8]. In this study, trust refers to the data sharing processes; in other words, trust in the sharing preferences of the data entered in the app by farmers. A recent study explored whether farmers are interested in joining a big data platform and, if so, what elements of the platform such as financial utilities would maximise their participation [9,10]. It was found that relatively small financial and nonfinancial benefits increased participation, even among farmers who stated strong privacy preferences [9]. Financial incentives refer to monetary benefits that are offered to farmers to encourage certain behaviour, changes, or actions. Partly covering data connection cost or payments for data sharing can act as financial incentive to stimulate famers' uptake or to remove barriers. A review of digital applications for agricultural purposes suggested that data privacy and incentives might be factors preventing farmers' participation [11]. However, knowledge is still limited about the incentives and benefits expected by farmers from mobile phone services [3,12]. Therefore, it is relevant to study farmers' preferences on who accesses their data and farmers' motivation to then explore feasible and fair nonfinancial benefits that might stimulate their uptake of digital solutions and participation.

Discrete choice experiments (DCEs) are suitable methods to elicit preferences for product characteristics when a product is new or not yet commercially available [13]. DCEs are gaining interest in agricultural research and have been used recently, for example, to study preferences regarding interactions with markets with focus on certification or contracts [14–16] or preferences for crop characteristics [17–19] or farming practices such as the use of inputs such as fertilisers, pesticides, or the application of conservation measures [20–24] and also to investigate preferences regarding agricultural or environmental policies [25–28].

Thus far, few studies have used choice experiments to study famers' preferences for ICT-based extension tools: Altobelli et al. [29,30] considered irrigation advisory services in Italy, Oyakhilomen et al. [31,32] studied ICT-enabled extension services, and Tesfaye et al. [33] studied climate services in Ethiopia. These studies, however, mostly focus on the preferred content rather than on the characteristics of the interaction with the ICT tool.

Receiving agronomic advice, capacity building, and seed innovation have been reported as examples of most needed incentives [34]. Even if an app is technically feasible and provides tailor-made and timely advice, farmers in constraint-based regions might require nonfinancial incentives to use it [34]. Hence, advancing the knowledge on farmers' motivations and preferences for nonmonetary incentives could help to increase smallholder farmers' intention to use apps, especially in constraint-based contexts, where financial incentives are often not an option. This is particularly the case with apps being developed by research-for-development agencies or institutes working in understudied geographical areas such as Central and South America. The potential for agricultural apps is large in the study location because of its high mobile phone adoption rates (by 2025, at least 75% of the population in Mexico will adopt and use a smartphone [35]). Moreover, in Mexico, ICT innovations in the agricultural sector are gaining attention from government agencies supporting the development of mobile phone apps to connect farmers with buyers or to deliver advice on crop production to B-corporations with, for example, the "Extensio platform" (previously Esoko), which provides content to Mexican farmers through SMS, a call centre, and a smartphone app. Against this background, in an earlier study, the drivers of the intention to adopt agricultural advisory apps in general were investigated

using a structural equation model [36]. The focus of the present study is on the use of a newly developed app called AgroTutor [37], which is developed for extension in Mexico. A DCE is applied to analyse farmers' preferences, as prospective users, for data sharing and nonfinancial incentives. The current study thus can provide insights to support developers of decision support apps as well as research-for-development practitioners and decision makers in the field of digitisation in agriculture. Specifically, this study looks at the importance of providing access to training and getting support during first-time use. It also considers whether farmers are open to share their farm data with other farmers, government and research institutes, and private companies. By applying a latent class (LC) model, heterogeneity within the farming population is also considered to support better targeting.

## 2. Research Context and Methods

### 2.1. The AgroTutor App

Created in 2017 and currently in the second phase of development along with the International Institute of Applied Systems Analysis (IIASA-Austria), the AgroTutor mobile phone application is a pilot project of the International Maize and Wheat Improvement Center (CIMMYT) that is being tested in the state of Guanajuato in Central Mexico. The smartphone app provides farmers with access to best practices and geo-referenced, timely information about fields and crops, including benchmarking data for crop placement, agronomical recommendations (i.e., optimising use of fertilisers), potential yield and financial benchmarking information (i.e., prices and costs), historical and forecasted weather data, and other expert sources of agricultural information in the region [37]. Farmers can also provide their own information regarding soils, management, and yields for use in crop models and for generating improved recommendations [37].

### 2.2. Models' Description

Discrete choice experiments allow the ex ante determination of preferences [38]; they present respondents with a choice between two or more alternatives described by pre-established attributes. In this case, attributes simulate different configurations which the AgroTutor app could take when launched. The method is in accordance with the theory of random utility and the Lancaster [39] attribute theory of value, which states that a good can be described as consisting of a bundle of characteristics at certain levels. It states that utility is not derived from the good as such, but rather from the specific attributes. The theory of random utility elucidates that when presented with two or more options, people decide in favour of the one option providing them with highest utility. Two models were applied and described in the following section.

#### 2.2.1. Conditional Logit

In a first step, a conditional logit was used to analyse the key attributes. Conditional logit analysis is the traditional model for the analysis of DCEs, explaining the preference of individuals based on the attributes in the choice cards and assuming homogeneous preferences among respondents [40]. Formally, as proposed by McFadden [41], the utility that each individual or respondent obtained from an alternative is the sum of utility from the individual characteristics:

$$U_{ij} = \beta' x_{ij} + \varepsilon_{ij} \tag{1}$$

The probability of choosing one alternative over another depends on the value of the utility. When the utility of alternative i is greater than the utility assigned to other alternatives, the alternative i will be chosen:

$$\text{Prob}_i = \text{Prob}\,(U_i \geq U_j)\ \forall\ j \in j = 1, J; i = j \tag{2}$$

### 2.2.2. Latent Class Model

In a second step, we explored the heterogeneity between groups of respondents using an LC analysis. The LC model captures the heterogeneity of respondents by dividing respondents into different groups or classes [42]. The number of classes is determined endogenously. Respondents are assigned to an LC or group with homogeneous preferences, but the observations belonging to each group are not revealed to the analyst. In this paper, we used a standard LC model specification [43]. Thus, the utility function of each individual i that belongs to an LC c is:

$$U_{jit|c} = \beta'_c x_{jit} + \varepsilon_{jit} \tag{3}$$

where the utility is a function of $\beta$ parameter estimates and the attributes composing the alternative. For each class, class-specific parameters $\beta_c$ will be estimated, together with estimations for each individual a set of probabilities of belonging to a certain class. In each class, the choice probabilities are defined as:

$$\text{Prob}[y_{it} = j|\text{class} = c] = \frac{\exp(\beta'_c x_{jit})}{\sum_{j=1}^{J_i} \exp(\beta'_c x_{jit})} \tag{4}$$

Presence in a determined class with specific preferences is probabilistic. The probabilities can be specified in the function of individual characteristics $z_i$, such as economic and attitudinal characteristics of the respondents. Then, the class probabilities will be a function of class parameters $\theta_c$, in respect to a reference class:

$$\text{Prob}[\text{class} = c] = \frac{\exp(\theta'_c z_i)}{\sum_{j=1}^{J_i} \exp(\theta'_c z_i)} \tag{5}$$

The unconditional probability that any randomly selected respondent chooses an alternative is obtained by combining the conditional probability in (4) with the class membership probability in (5), resulting in the following equation:

$$\text{Prob}(y_{it} = j) = \sum_{c=1}^{C} \text{Prob}(\text{class} = c) \frac{\exp(\beta'_c x_{jit})}{\sum_{j=1}^{J_i} \exp(\beta'_c x_{jit})} \tag{6}$$

The optimal number of classes is determined based on the pseudo $R^2$, Akaike Information Criterion, and the Bayesian Information Criterion [44,45]. The LC model provides evidence for systematic heterogeneity in the preference structure of farmers. To estimate this heterogeneity, the LC model was run several times with an increasing number of classes and different combinations of class membership variables.

### 2.2.3. Attributes and Levels

A standard set of stages were followed to design the DCE [38]. The first stage was to select relevant attributes for the usage of an app. The attributes were identified based on a literature review and consultation with CIMMYT experts, individual interviews with extension agents and farmers, and a participatory workshop with both extension agents and farmers in the region (Table 1). The attribute characteristics were related to the different requirements of farmers for using this app. The second stage aimed to assign key attribute levels. Even though there is no common agreement on the number of levels, levels should reflect a realistic and feasible scenario [13]. Six attributes were identified at this stage (Table 1). A detailed description and references used can be found in Appendix A.

**Table 1.** Attributes and levels for the choice experiment on an app for farmers.

| Attribute | Definitions | Attribute Levels | Comments |
|---|---|---|---|
| Support during first-time use | Whether farmers use the app by themselves or with an extension agent's help | Get support from an extensionist to do it yourself | "Input time" attribute considers time spent learning how to use the app. |
| Data input requirements | How often farmers are expected to enter or update information | No requirement<br>Once every 2 weeks<br>Once every 2 months<br>Once every production cycle | If farmers perceive minimum required as easy to comply with, it may incentivise participation. |
| Data-usage cost | Cost associated with the internet data spent on accessing and conducting basic tasks in the app (each time app is accessed). | 0 MXN<br>5 MXN<br>10 MXN | The 0 MNX level is an option to offer offline features too. The cost–benefit perceived by the farmers may be a factor or barrier to farmer participation or continuous usage. |
| Access to training | Special access to training and capacity building events in their region, in exchange for using the app. A nonfinancial incentive. | Special access<br>No special access | Allows one to explore whether farmers will accept training and capacity building as compensation (nonfinancial utility) and whether this will motivate farmers' preference for the use of the mobile phone app. |
| Access to shared data | To which extent the data recorded in the app are accessible to others apart from the user farmer. | Only me<br>All other farmers<br>Research institutions and government<br>Private companies | Research institutes and government combined due to the nature of the case study in which the institute developing the app works closely with the government in the region. |
| Replacing extension service visits | Whether the farmer prefers to keep (or not) the extension services visits. | Extension service continues to visit<br>No regular extension service visits | Important to examine whether the app might replace the visits. |

The third stage in the DCE methodology is designing the choice set. A choice set is a group of hypothetical alternatives constructed through experimental design. The design is a D-efficient design, assuming priors for the different attributes according to theoretical expectations from the literature, which generates a sample of the full design in such a way that the most important effects can be estimated [13,46]. While it is then recommended to run a pre-test of the choice experiment and use the first results as priors to improve the design, the planning of the data collection did not allow for this. The full factorial design consists of $2^3 \, 4^2 \, 3^1$ (=384) alternatives. Representing all possible combinations of these scenarios would be unfeasible for respondents. Therefore, a statistically efficient choice design combining the attribute levels into alternatives and choice sets was constructed using Ngene [47]. The design was estimated using the expected signs of the attributes based on the literature and expert consultation as priors. Negative preference was assumed for more data input requirements, higher costs of the usage, and more widely sharing the data, while a positive preference was assumed for incentives such as special training, help during first-time use, and visits of the extension service. An efficient design consisting of 24 alternatives was obtained and arranged into 12 choice sets, where two alternatives were compared. These were then assigned into two blocks of six choice sets to avoid a survey that was too lengthy (Figure 1). Respondents were then randomly allocated to the two blocks. For each choice set, the respondents were asked to choose between one of the two profiles of app usage. The design also included an opt-out option so as not to force the respondents to choose one of the alternatives when those were not considered suitable. The opt-out was described as "under these conditions I prefer none of the described alternatives".

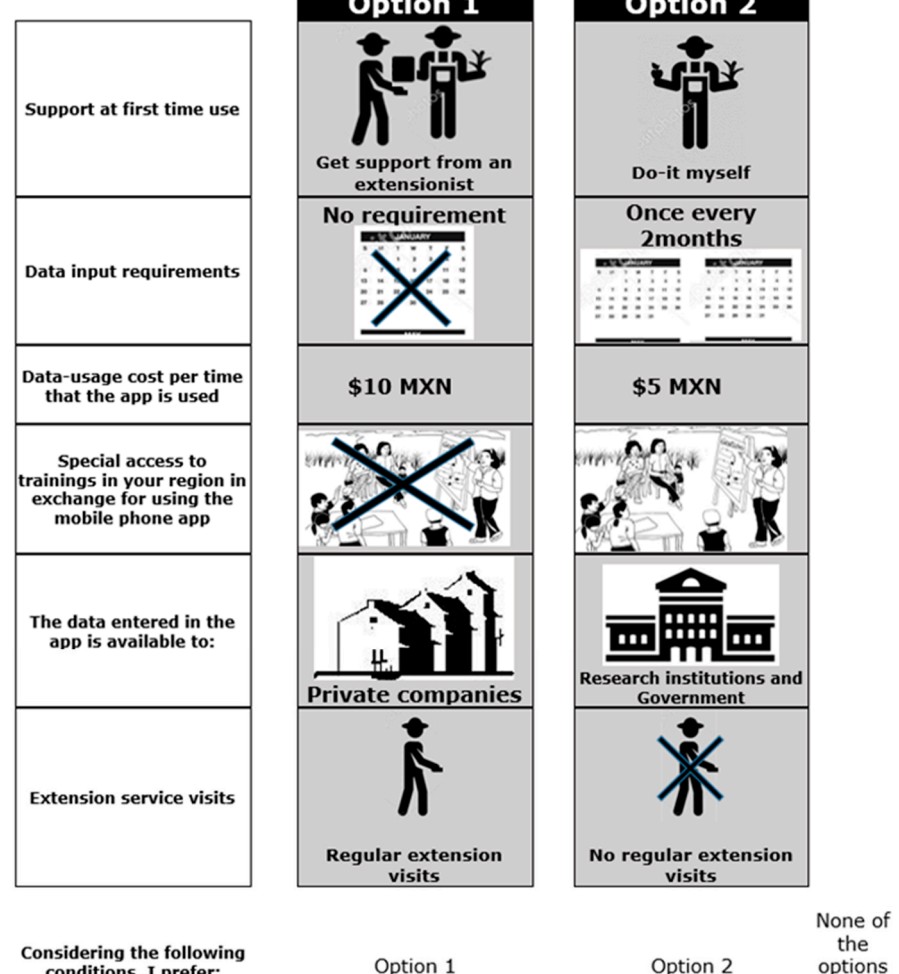

**Figure 1.** Example of a choice card.

### 2.3. Data Collection and Analysis

A survey was used to investigate attitudes towards the use of the AgroTutor app and was conducted in the context of the CIMMYT innovation hub in Guanajuato, Mexico. The innovation hub model comprises research platforms, demonstration modules, and extension and impact areas [48]. In research platforms, local researchers adapt farming innovations to their specific conditions. The demonstration modules are on farmers' land and involve side-by-side comparisons of new technologies and conventional practices. Module outcomes often generate feedback to research platforms and allow for farmer-to-farmer interaction and sharing. Around this physical infrastructure of the hub, a network of actors in the value chain is built with the objective to drive adoption, local impacts, and the scaling of innovations.

The survey had two main parts: a part collecting data on socio-economic and farm-related characteristics, and secondly, the DCE. Different determinants of behavioural intention to adopt were measured and applied as grouping variables from the Unified Theory of Acceptance and Use of Technology [49], and the orientation goal theory [1].

A description of the constructs can be found in Molina Maturano et al. [36]. Data were collected through face-to-face interviews by a group of enumerators. To reduce the bias of the respondents, extensive training was provided to ensure that all enumerators conducted the survey in a similar way and to assure the respondents that their privacy would be kept. Given that the app was new to all respondents, a video describing the content and capabilities of the app was shown to all respondents before they answered the choice experiment. In this way, it was ensured that all respondents had similar information

and knowledge of the app under consideration. We collected 392 valid surveys, both from respondents that were connected (204) and nonconnected (188) to the innovation hub. The sampling procedure is described in Molina Maturano et al. [36]. The socio-economic and farm characteristics were evaluated using descriptive statistics in SPSS. The models described were estimated using NLogit 5 software. To estimate heterogeneity, the LC model was run several times with increasing numbers of classes and different combinations of the above-mentioned class membership variables. Latent class models attempt to capture heterogeneity of the respondents depending not only on the socio-economic characteristics of the respondents, but also on attitudes and values of the respondents. In this regard, preference formation might be affected by factors that are not directly observable, such as the intention to adopt an agricultural app.

Three components obtained from a Structural Equation Model (SEM), which satisfied the criteria of significantly influencing the behavioural intention, were explored as membership and profiling variables: the performance expectancy (PE), facilitating conditions (FCs), and mastery approach goals (MAGs) [36]. A two-step estimation was used to include these constructs as latent variables. First, the SEM was estimated, and factors from the resulting model were retrieved. Once the model was estimated, categorical variables were created based on the obtained scores and used as explanatory variables of the LC model. In other words, the categorical variable separates participants in having higher or lower levels of PE, FCs, and MAGs. Such a use of psychometric indicators such as attitudes to construct LCs in choice models has recently gained popularity [50,51].

The farmers' segments obtained through the LC modelling were further profiled using the abovementioned profiling variables. To gain an overview of the characteristics of the classes, post-estimation analyses were performed to test whether significant differences existed between the obtained classes. Means were compared through a one-way ANOVA analysis applied when necessary. Categorical variables were analysed using a chi-square association test, while noncategorical variables were analysed using an independent-samples Kruskal–Wallis test. For simplification purposes, noncategorical variables resulting non-significant were grouped based on the mean and converted to categorical variables. In a first step, farmers' classes were compared to test whether overall significant differences exist across the segments at a significance level of $\alpha = 0.05$. If overall significant differences were found, pair-wise comparisons were performed to identify which consumer segments differed significantly based on the adjusted significance.

## 3. Results and Discussion

### 3.1. Descriptive Statistic

The average age of the respondents was 55 years (Table 2). This is consistent with reported average farmers' ages of 54.6 years in Mexico [52] but is higher than in another DCE study on ICT-based advisory tools in Nigeria, in which the average age was 44 years [32]. Overall, crop production was identified as the main activity by the respondents. About 85% of the respondents were smallholders who own land, with 70% having an agricultural contract for production. One out of three respondents received advice from an extension agent every week, while 19% were not receiving any advice from extension agents (Table 2). This shows that the frequency of advice is not limited for at least one third of the sample; therefore, certain aversion to the replace extension services is expected. In total, 82% of the respondents owned a mobile phone (all types) and used the phone, on average, for 43 min a day. These habits provide insights on the communication channels currently being used by smallholders in the region that are not necessarily being applied extensively yet for agricultural purposes. Moreover, the farmers spent 170 MNX (8 EUR) per month on mobile phone credit, which can be used for calls, SMS, and internet, which provides a baseline for exploring further financial incentives, if available, to promote the actual usage of an app.

**Table 2.** Socio-economic, land characteristics, innovation hub, and mobile phone usage.

| Parameters | Total Sample (392) | Parameters | Total Sample (392) |
|---|---|---|---|
| Socio-economic: | | Connection to the innovation hub | |
| Age (average; range) | 55; 23–86 | % farmers linked to the hub | 52% |
| Gender (% male/female) | 94%/6% | Time linked to the hub (years) | 3 |
| Sample share living w/youth * | 67% | Extension services: | |
| Literacy ** | 95% | % of farmers receiving weekly advice | 32% |
| Crop production as primary income source (%) | 91% | Mobile ownership and use: | |
| Land characteristics: | | % owning a mobile (all types) | 82% |
| Land area (average ha; range) | 16.5; 2–600 | % owning a smartphone | 46% |
| % with a production contract | 74% | Time using a smartphone (years) | 3.5 |
| % owning land | 75% | Time using the phone (min/day) | 43 |
| Area of land owned (average ha) | 9 | Mobile credit spent (MXN/month) | 170 *** |

* 18–35 years old living at home, ** read and write, *** around 8 EUR based on 2019 exchange rate.

Female respondents only represented 6% of the total sample. This is also consistent with the latest Agricultural National Survey, in which only 14% of producers and decision makers of the production unit, at a country level, were women [53]. Previous studies in rural Indian contexts have shown that women farmers value mobile-enabled services to increase their knowledge of climate-smart technologies and encourage their participation in decision making [54,55]. The limited sample size did not allow us to conduct further analysis based on gender, but it is suggested that further studies focus on gender-inclusive solutions and preferences.

*3.2. Farmers Preferences: Conditional Logit Model Results*

Across the sample, significant estimates of nonfinancial incentive(s) were found in the CL model for: support during first-time use, special access to training in the region, and data-usage cost (Table 3). Farmers showed a strong positive preference for both support at first use and special access to training. The first is likely to be related to the fact that only 46% own a smartphone or due to farmers' average age and subsequent lack of digital literacy. Therefore, feasible options to provide this type of support during the launch could be extension services or youth living with the farmers (end-users), since more than half of the respondents lived with youth members between 18 and 35 years old who might have better digital literacy. This could also be a pathway to engage youth in agricultural activities, as farmers' ageing and lack of generational relief is a current challenge in Mexico [52] and worldwide.

Both positive preferences suggested that nonfinancial incentives related with capacity building might stimulate uptake of the AgriTutor app. For this, the extension services play a crucial role [56]. Because nearly half of the respondents are linked to an innovation hub that provides extension services (among other services), the preference found to keep extension services is not surprising. Moreover, it is in line with previous studies, where extension services have proven to be one of the most trusted sources for agricultural advice by smallholder farmers [57,58]. Similarly, the role of advisors in helping farmers to create more value out of smart agriculture tools has already been identified in the previous literature in developed regions [56]. The preferences of connected farmers towards extension services provision also suggest that replacing extension services completely with the mobile phone app might be contra productive.

A negative preference for the cost of data usage was found, as expected, suggesting that if the cost paid for accessing basic app features increases, the usage would decrease. The average spent amount found among respondents (170 MXN/month) is comparable with the estimated cost to be paid for accessing basic app features (150 MNX/month). It means using the app daily for a month to conduct basic functions. Although this study does not cover a willingness to pay nor focuses on financial incentives, it provides insights of a cost baseline for further exploration of financial incentives. A significant negative preference was found for registering data minimum every two weeks (versus 'no requirement' of data

needed to be registered), but only for the nonconnected farmers. For farmers connected to the innovation hub, a negative preference was found for data sharing with private companies, as compared to limiting access to farmers. These results place into question the notion of self-evident trust relationships within an innovation hub. This finding supports previous results about farmers' concerns of data privacy [59] and the importance of not assuming their acceptance, especially when external or private actors enter the scene. This is also highlighted by the World Bank [8]. Regardless of the connection of the farmer to the innovation hub, a preference was found for the data-usage cost (negative) and training (in exchange for using the app) (positive). The integration of these positive incentives during the app launch and scaling-up stage could incentivise the initial adoption and sustainable use of this type of apps.

**Table 3.** Conditional logit model estimates.

| Number of Respondents | All Farmers | | | Connected Farmers | | | Non-Connected Farmers | | |
|---|---|---|---|---|---|---|---|---|---|
| | 392 | | | 204 | | | 188 | | |
| | Estimates | | Std. error | Estimates | | Std. error | Estimates | | Std. error |
| Support at 1st time use | −0.314 | *** | (0.093) | 0.347 | *** | (0.133) | 0.299 | ** | (0.131) |
| Data input requirements ° | | | | | | | | | |
| Once every 2 weeks | −0.114 | NS | (0.099) | 0.043 | NS | (0.144) | −0.270 | ** | (0.137) |
| Once every 2 months | −0.058 | NS | (0.108) | 0.047 | NS | (0.160) | −0.160 | NS | (0.148) |
| Once every productive cycle | 0.105 | NS | (0.142) | 0.303 | NS | (0.207) | −0.089 | NS | (0.197) |
| Data-usage cost | −0.046 | *** | (0.008) | −0.066 | *** | (0.012) | −0.027 | *** | (0.012) |
| Access to trainings | 0.310 | *** | (0.068) | 0.298 | *** | (0.096) | 0.332 | *** | (0.098) |
| Access to shared data + | | | | | | | | | |
| All | −0.025 | NS | (0.103) | −0.074 | NS | (0.147) | 0.014 | NS | (0.144) |
| Research institutes and government | −0.061 | NS | (0.087) | −0.113 | NS | (0.130) | −0.042 | NS | (0.120) |
| Private companies | −0.213 | ** | (0.085) | −0.290 | ** | (0.124) | −0.136 | NS | (0.119) |
| Replacing extension service visits | 0.298 | *** | (0.063) | 0.520 | *** | (0.091) | 0.074 | NS | (0.090) |
| ASC + | −0.869 | *** | (0.105) | −0.827 | *** | (0.152) | −0.896 | *** | (0.147) |

Note: *** and ** indicate significance at 1% and 5% level, respectively. NS indicates no significant effect. Standard errors are in brackets. ° versus no requirement of minimum data input needed to be entered by the use, + versus farmer only. ASC: alternative specific constant. + The alternative specific constant (ASC) is a dummy variable attached to the opt-out option in the choice cards. It is a dummy variable with the value 1 associated with the choice for the 3rd alternative (opt-out) and a 0 when alternatives 1 and 2 are chosen.

*3.3. Farmers Valuing Extension Services, Data-Usage Cost and Data Privacy: Latent Class Model Results*

The regression results (estimates) of the LC model with three classes resulted from a model with classes sizes representing at least 10% of respondents (Table 4). The segments (classes) were further profiled based on socio-economic characteristics, extension services, and mobile phone habits (Table 5). The class assignment coefficients reflect the effects of the following retained variables: being linked to an innovation hub, farmers' age, and mobile phone ownership, on the individual's class assignment with class 3 as a reference group. Therefore, statistically significant, positive coefficients for class assignment always indicate that a farmer is more (or less) likely to belong to the respective class than belonging to class 3. The opt-out alternative was chosen 14.7% of the time.

The segment class 2, labelled as 'value mastery, cost-averse', represents 11% of the surveyed farmers with a stronger negative preference for the cost associated with the use of the app, as compared to class 1 and 3 farmers. These results are in line with Yigezu et al. [60], who argued that farmers view innovations as potential risks rather than opportunities in their study on the low and gradual adoption of costly practices by smallholders. However, field days, demonstration trials, and free access to equipment for first-time users increase the adoption [60]. Therefore, special access to training could help to engage this minority that is cost-averse.

**Table 4.** Regression results (estimates) of the latent class model with 3 classes.

| | Latent Classes | | | | | |
|---|---|---|---|---|---|---|
| | **Class 1 (50% (n = 197))** **Value Extension** | **Std. Error** | **Class 2 (11% (n = 45))** **Value Learning, Cost-Averse** | **Std. Error** | **Class 3 (39% n = 151)** **Value Data Privacy** | **Std. Error** |
| | Preference parameters | | | | | |
| Support at 1st time use | 1.4547 *** | 0.4523 | 0.4060 | 0.5582 | 0.0566 | 0.1969 |
| Data input requirements ° | | | | | | |
| once every 2 weeks | 0.4726 | 0.3276 | 0.0241 | 0.7752 | 0.5280 *** | 0.1777 |
| once every 2 months | 0.1944 | 0.3614 | 0.1537 | 0.7458 | 0.3879 ** | 0.1868 |
| once every cycle | 0.0653 | 0.6470 | 0.6984 | 0.9075 | 0.1627 | 0.2758 |
| Data-usage cost | 0.0644 * | 0.0359 | 0.1659 *** | 0.0567 | 0.0210 | 0.0168 |
| Access to trainings | 1.4137 *** | 0.3590 | 0.7436 * | 0.4044 | 0.0883 | 0.1360 |
| Access to shared data + | | | | | | |
| All | 0.3464 | 0.3460 | 0.7116 | 0.5665 | 0.1134 | 0.1997 |
| Research institutes and government | 0.3114 | 0.3104 | 0.5013 | 0.5865 | 0.0923 | 0.1401 |
| Private companies | 0.4392 | 0.2855 | 0.0467 | 0.6283 | 0.6382 *** | 0.1683 |
| Extension services | 1.5826 *** | 0.3470 | 0.3974 | 0.4757 | 0.0218 | 0.1255 |
| ASC | 0.0999 | 0.4809 | 2.6980 *** | 0.7900 | 2.3486 *** | 0.2473 |
| | Class assignment | | | | | |
| Constant | 0.5196 | 0.4244 | 1.4032 ** | 0.6152 | | |
| Age (1 if > 55) | 0.2686 | 0.3142 | 0.2173 | 0.4700 | | |
| Own mobile (Yes, No) | 0.0437 | 0.4082 | 0.6323 | 0.5639 | | |
| Linked to innovation hub (Yes, No) | 0.9523 *** | 0.3053 | 0.5841 | 0.5783 | | |
| Behavioural intention (1 if > median) | 0.3485 | 0.4242 | 1.8744 *** | 0.6800 | | |
| Mastery approach goal (1 if > median) | 0.6493 | 0.4262 | 0.9156 * | 0.5321 | | |

Note: ***, **, and * indicate significance at 1%, 5%, and 10% level, respectively. ° versus no requirement of minimum data input needed to be entered by the use, + versus farmer only.

**Table 5.** Profiling farmers' latent classes.

| Variable | Levels | Total Sample | Class 1 (50%) Value Extension | Class 2 (11%) Value Mastery Cost-Averse | Class 3 (39%) Value Data Privacy | *p*-Value |
|---|---|---|---|---|---|---|
| | Socio-economic: | | | | | |
| Age | $\mu$ ($\pm\sigma$) | | 53 (14.6) | 57.5 (12.8) | 57 (12.7) | 0.044 [3] |
| | none | 12% | 22 (11%) | 5 (11%) | 20 (13%) | |
| | elementary | 36% | 68 (35%) | 16 (36%) | 56 (37%) | |
| Education level | high school | 33% | 63 (32%) | 17 (38%) | 49 (32%) | 0.810 [2] |
| | university | 19% | 44 (22%) | 6 (13%) | 26 (17%) | |
| | undergraduate | 5% | 8 (4%) | 3 (6%) | 9 (6%) | |
| Land area | $\mu$ mean ($\pm\sigma$) | | 15 (28) | 18 (54) | 9 (13) | (1–3) 0.002 [2] |
| Land area owned | | | 9 (28) | 17 (55) | 7 (14) | (2–3)0.031 [2] |
| | Extension services: | | | | | |
| Connected to the innovation hub | No | 48% | 78 (40%) [a] | 23 (51%) [a,b] | 88 (58%) [b] | 0.002 [1] |
| | Yes | 52% | 119 (60%) [a] | 21 (49%) [a,b] | 63 (32%) [b] | |
| Years linked to the hub | $\mu$ mean ($\pm\sigma$) | | 2.9 (2) | 3.5 (1.8) | 2.5 (1.3) | 0.091 [3] |
| | Never | 19.4% | 28 (14%) [a] | 14 (32%) [b] | 34 (23%) [a,b] | |
| | every 6 months | 13.3% | 20 (10%) [a] | 4 (9%) [a] | 28 (19%) [a] | |
| Frequency of advice received | every 2 months | 7.4% | 14 (7%) [a] | 1 (2%) [a] | 14(9%) [a] | 0.002 [1] |
| | every month | 23% | 49 (25%) [a] | 9 (20%) [a] | 32 (21%) [a] | |
| | every week | 32.1% | 74 (38%) [a] | 16 (36%) [a,b] | 36 (24%) [b] | |
| | every 3 days | 4.8% | 12 (6%) [a] | 0 (0%) [a] | 7 (5%) [a] | |
| | Mobile phone services: | | | | | |
| Own mobile | No | 17.9% | 31 (16%) [a] | 13 (30%) [b] | 26 (17%) [a,b] | 0.093 [1] |
| | Yes | 82.1% | 166 (84%) [a] | 31 (70%) [b] | 125 (83%) [a,b] | |
| | None | 17.9% | 31 (16%) [a] | 13 (30%) [a] | 26 (17%) [a] | |
| Mobile type | Smartphone | 45.9% | 111 (56%) [a] | 15 (34%) [b] | 54 (36%) [b] | <0.001 [1] |
| | Basic | 19.4% | 26 (13%) [a] | 12 (27%) [a,b] | 38 (25%) [b] | |
| | Medium | 16.8% | 29 (15%) [a] | 4 (9%) [a] | 33 (22%) [a] | |
| Years using a smartphone | $\mu$ ($\pm\sigma$) | | 3.9 (2.9) | 3.2 (2.5) | 3.2 (2.7) | 0.262 [3] |
| Minutes per day | $\mu$ ($\pm\sigma$) | | 52 (71) | 36.8 (56) | 32.5 (68) | (1–3) 001 [2] |
| Spend in mobile credit/month | $\mu$ ($\pm\sigma$) | | 181.6 (97.5) | 153 (66.3) | 161.4 (89.8) | (1–3).047 [2] |

[1] Chi-square, [2] Kruskal–Wallis, [3] ANOVA. Each subscript letter [a,b] denotes a subset of categories whose column proportions do not differ significantly from each other at the 0.05 level.

The mastery approach goals (MAGs) and behavioural intention (BI) retained variables, obtained from the SEM, were also used as class membership parameters in the LC model (Table 4). MAGs refer to the intention to understand something new or to improve the level of competence [61] and were chosen due to the link with the DCE attributes of support for the first-time use and especial access to training. MAGs are also critical to be studied along with farmers' age, especially in this case, in which farmer mean age was 55 years (Table 2) and their openness to new technologies might have been less than younger farmers. The MAGs, studied as a dummy variable, suggested that individuals with mastery goal orientation are more likely to develop a higher sense of confidence [1,61]. Surprisingly, having higher mastery approach goals further increases the probability of belonging to this class, while lower behavioural intention to adopt the app decreases the probability of belonging to class 2. This finding seems to contradict the idea that mastery approach goal is a driver for the intention to adopt.

One possible reason for this divergence could be explained by the class negative preference for the associated cost (Table 4). This means that even though farmers in this class are willing to learn and master an app, they are less likely to adopt the app because of affordability reasons. Another possible explanation is that because of their age, farmers prefer in-person training instead of completely relying on the app advice. This suggests that nonfinancial incentives such as training might be a way to promote the willingness to adopt the app in this group. This finding is also in line with the conclusion of Baumüller [12] about making mobile phone services to farmers more appealing if the complexity of services is handled by the service provider or intermediaries such as extension agents, among others.

One more class assignment variable corresponded to belonging to the innovation hub, in which farmers are participants of new practices and innovations, especially around sustainable agriculture in the region. The LC model results demonstrate that farmers connected to the innovation hub are more likely to belong to class 1. Farmers in class 1 have land sizes of 15 ha on average, which is significantly larger than for farmers in the reference class (class 3). As for the mobile phone characteristics, 25% of the farmers owning a smartphone belong to class 1, and they have been using a mobile phone for more than 2 years. It also appears that farmers belonging to this class are more likely to have a higher phone usage (min per day) than class 3. This finding could explain why the farmers that are more likely to belong to this class do not have a negative preference towards the frequency of entering data in the app, nor do they have a strong preference for data sharing. The farmers segment (class 1) labelled as 'value extension services' represents 50% of all farmers and are shown to significantly prefer to keep extension services visits and have special access to training for using it. This finding is consistent with the profiling results (Table 5), showing that most of the farmers belonging to class 1 'value extension' frequently received advice. This finding is also in line with previous studies on the role of extension services and advisers for making sense of the obtained results or advice from digital tools [56,62]. Additionally, it is in line with the development, by public organisations, of training programs, including the support of initiatives such as farmer clubs [63] for the uptake, diffusion, and scaling of these innovations. These insights are of special interest whenever these applications are meant to be scaled and when private and external partners will be involved in the innovation system for effective adaptation from the diverse digital innovation practices of advisors [62].

Farmers in class 3, labelled as 'value data privacy', represent 39% of the surveyed farmers. Contrary to farmers in the 'value extension services' class, these farmers have a negative preference for updating information in the app frequently (once every 2 weeks and 2 months) in comparison to 'no requirement' attribute for registering information. In addition, they also have a negative preference for sharing data with private companies (versus sharing only with themselves and their peers). These findings are in line with recent studies on the willingness of farmers to give and share data [9], which further shows that the organisation operating the platform, in this case the app, is particularly important:

farmers are most willing to share their data with universities and researchers and least willing to share their data with the government.

## 4. Limitations

Despite its contributions regarding farmers' preferences to use the AgroTutor app that provide agricultural-related information about crops, this study has some limitations. First, the preferences might differ from location to location, so comparing farmers across different cultures both in developed and developing countries would be theoretically and practically useful to further validate the outcomes of this model. Our study does not claim to statistically represent farmers in the whole country (e.g., in terms of gender or location). Second, as the findings apply to a specific constraint-based context, care must be taken when interpreting the findings and aiming to generalise to other geographies with other ICT infrastructures. Further validation needs to be extended to other geographical regions within Mexico and Central and South America. Future willingness-to-pay studies could also verify whether the preferences are robust over time. From a methodological standpoint, further bias checks are recommended, such as a conventional attribute non-attendance model (conventional ANA) and a validation attribute nonattendance model (validation ANA) that implies non-compensatory decision-making behaviour of respondents [32]. This means a correction for farmers that may not make an expected trade-off between all attributes of the various alternatives.

## 5. Conclusions and Implications

In an ex ante, discrete choice experiment, this article analysed the effect of data-sharing rules and nonfinancial incentives on the use of AgroTutor, an agricultural advisory app. For this, 392 Mexican farmers were surveyed. The results show that providing access to special trainings would clearly increase use of the app, while a configuration in which app-related data would be shared with private companies was negatively perceived by 44% of the respondents. In general, farmers, despite their age, seemed to support the use of ICT-based, site-specific extension services. These findings call for implementers to contribute to the responsible implementation of such apps, while considering smallholders' privacy and data ownership.

Results from the LC model demonstrate differences in preferences when farmers' connectedness to the CIMMYT innovation hub and mastery approach goals variables are considered as a grouping variable. These variables have an effect on farmer preferences for data sharing. Three different farmer segments were distinguished; those connected to an innovation hub are most likely part of the group of 'value extension' farmers (45%). They are also currently receiving extension services more frequently and have less aversion against data sharing. The second segment comprises the 'value mastery, cost-averse' (11%) group; they had a lower behavioural intention to adopt and had a lower smart phone ownership, while the 'value data privacy' (44%) group finally had a clear negative preference to share their data with private companies. These farmers typically have smaller landholdings, receive less extension services, and are less likely to be connected to the innovation hub. App developers and implementers can use such information to close the digital divide in a targeted way by providing diverse incentives and tailoring ICT-based technologies. This is in line with the recommendations by the World Bank [8], where one of the lessons provided is that the focus should be on the demands and needs of the target population, which can be heterogeneous.

Furthermore, the preferences of farmers connected to an innovation hub with regard to extension suggest that replacing completely extension services with the mobile phone app will not be perceived as ideal. Combinations with traditional forms of communication and knowledge sharing are also recommended by World Bank [8]. Our study shows that traditional extension services and training could even act as nonfinancial incentives to use the app. This suggests the importance of flexible extension systems that consider farmers' preferences of usage and their trust in the different actors involved in sharing data and

information via a mobile phone app, and correctly inform farmers not only about technical risks of advice (yield and returns), but also on data ownership and privacy.

**Author Contributions:** Conceptualisation, J.M.-M., J.T.-C., A.G.-M., D.T.G., S.S.; data curation, J.M.-M.; formal analysis, J.M.-M., J.T.-C.; funding acquisition, A.G.-M., S.S.; investigation, J.M.-M., N.V.; methodology, J.M.-M., J.T.-C., S.S.; project administration, J.M.-M., N.V.; resources, A.G.-M.; supervision, N.V., B.G., S.S.; writing—original draft, J.M.-M.; writing—review and editing, N.V., D.T.G., B.G., S.S., H.D.S. All authors have read and agreed to the published version of the manuscript.

**Funding:** The survey work was part of the projects "Cultivos para México/MasAgro Productor" and "MasAgro Guanajuato", made possible by the generous support of the government of Mexico through SADER and the State Government of Guanajuato through the SDAyR, and is part of the CGIAR Initiative Excellence in Agronomy. Any opinions, findings, conclusions, or recommendations expressed in this publication are those of the authors and do not necessarily reflect the view of the donors mentioned previously.

**Institutional Review Board Statement:** Ethical review and approval were waived for this study, because no sensitive, personal data or geo localisation were collected from individuals. Best practices in the collections, anonymous treatment, and processing of the data were followed according to Ghent University guidelines concerning ethics in research.

**Informed Consent Statement:** Informed verbal consent was obtained from all participants in this study. The option to withdraw the interview at any time was provided.

**Data Availability Statement:** The data presented in this study are openly available in Dataverse at https://hdl.handle.net/11529/10548640 (accessed on 21 January 2022).

**Acknowledgments:** The authors are grateful to CIMMYT staff for the opportunity given to the first author to conduct a research stage. Special thanks to the surveyed farmers for their participation and to the team from MasAgro Guanajuato (Erick, Amador and extensionists) for their help with the logistics.

**Conflicts of Interest:** The authors declare no conflict of interest. The funders had no role in the design of the study; in the collection, analyses, or interpretation of data; in the writing of the manuscript; or in the decision to publish the results.

## Appendix A. Explanation of Selected Attributes

- "Support during first-time use" refers to the support provided by an extension agent to the farmer to introduce the app and show how it works. A previous CE study investigated the "input time" attribute in the context of a decision support tool adoption by extension advisers (Kragt and Llewellyn, 2014). The "input time" attribute considers the time spent learning how to use the app. During the field work and workshop preparing this study, the interaction of extension agents with the farmers while introducing the app for the first time was observed as relevant. As well, this attribute is also important to gain insights about the trust in the provider channel. Two levels were suggested: no help and with extension agent help. Additionally, the attribute is consistent with the importance of the credibility of the provided information to the farmers or agricultural professionals in the context of crowdsourcing in agriculture (Minet et al., 2017).
- "Data input requirements" refers to the minimum required frequency with which a farmer needs to register or update information in the mobile phone app. The information required might be the fertiliser use, farm data (input quantities, yields, and cereal type), or the registration of a plot. This attribute was relevant to gain insights in the preferences to update information in the app. Four levels were identified: no requirement, once every 15 days (2 weeks), once every 2 months, and once every productive cycle. If the minimum required is perceived by the farmers as easy to comply with, it may incentivise participation or continuous usage.
- "Data-usage cost" is the cost of internet data-usage every time the farmer accesses the app. A similar attribute has been used previously to assess extension agents'

stated preferences for the cost of a pest management decision support tool (Kragt and Llewellyn, 2014). The cost–benefit perceived by the farmers may be a factor or barrier to farmer participation or continuous usage. The cost was calculated along with the app developers from IIASA considering the cost of megabytes of internet used per time conducting basic actions in the app (register a plot, consult fertilisation advice, weather, and benchmark information). Around 5 megabytes are used per time (1 megabyte = 0.98 MXN); therefore, three levels were proposed: 0 MXN, 5 MXN, and 10 MXN. The 0 MNX level could be considered as an option under which the app will offer free offline features too.

- "Access to training" refers to a nonfinancial utility or compensation for the use of the mobile phone application as special access to trainings and capacity building events in their region. The access to face-to-face knowledge exchanges might be perceived as an incentive for farmers to keep on using an app or provide data entries. Hence, it aims to explore whether some farmers will accept training and capacity building as compensation (nonfinancial utility) and if it will motivate farmers' preference for the use of the mobile phone app.

- "Data sharing" refers to which actors the farmer prefers to have access to the information registered in the app. Data ownership is an important issue in the context of the current data revolution and big data applications (Wolfert et al., 2017). Choice experiments have been used before to examine privacy trade-offs in smartphone applications (Savage and Waldman, 2015) and also to estimate the value which app users gave to their friends' information (Pu and Grossklags, 2017). However, this aspect was only recently explored, with farmers looking at their willingness to join a big data platform (Turland and Slade, 2020). In our study, four levels are proposed: only me, everyone including peers, research institutions and government, and private companies. Research institutes and government are together due to the nature of the case study in which the institute developing the app works closely with the government in the region (Section 3.1).

- "Replacing extension services visits" considers the scenarios in which the extension agents keep visiting the farmers or not. The attribute is important to add to have control on the perception that the app might replace the visits. This was a concern raised by the farmers connected to the innovation hub during the preparatory interviews and field work as the extension advisers' visits are already being conducted in the study area. Two levels are proposed: extension services keep on visiting and no regular extension service visits.

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
