# Peer review of "How to Make a Smartphone-Based App for Agricultural Advice Attractive: Insights from a Choice Experiment in Mexico"

_agronomy, doi:10.3390/agronomy12030691_

Round 1

Reviewer 1 Report

This study fills an important research gap. The methods are innovative and the the paper is well written. The intepretation of the results could be improved. Please see the specific comments in the attached file.

Reviewer 2 Report

The article in general is quite rigorous and appropriate for this journal.

For both the introduction and the conclusions and implications, it would be useful for the authors to contrast their work with the following World Bank report:

  • World Bank. 2017. ICT in Agriculture (Updated Edition) : Connecting Smallholders to Knowledge, Networks, and Institutions. Washington, DC: World Bank.

In particular, the suggestion is to elaborate in the conclusions on the implications of their study for understanding and improving ICT-farmer relations.

A more detailed explanation of how the sample was obtained and selected is missing.
A more up-to-date bibliography on methodology:

  • Hensher, D.A.; Rose, J.M.; Greene, W.H. (2015). Applied Choice Analysis. 2nd edition. Cambridge University Press. https://doi.org/10.1017/CBO9781316136232
  • Hensher, D. & Johnson, L. (2018). Applied discrete-choice modelling. London: Routledge.

It would be necessary to provide a link and a more detailed explanation with screenshots of how the app works.

https://apps.apple.com/es/app/agrotutor/id1457033299

It would be convenient to set out in a table 1 the definition and measurement of the variables with a brief explanation and measurement (response alternatives) and a descriptive such as the mean and standard deviation.
Furthermore, it is not necessary to explain in detail the formulas that explain and support the "Discrete Choice experiment" methodology, but it might be more appropriate to detail or review other articles that have applied this analysis technique to the field of agriculture or agronomy. Especially those researches and publications that have been published recently, such as for example:

  • Mazzocchi, C., Sali, G. (2022). Supporting mountain agriculture through “mountain product” label: a choice experiment approach. Environ Dev Sustain 24, 701–723  https://doi.org/10.1007/s10668-021-01464-3
  • Wieczerak, T., Lal, P., Witherell, B. et al. (2022). Public preferences for green infrastructure improvements in Northern New Jersey: a discrete choice experiment approach. SN Soc Sci 2, 15 https://doi.org/10.1007/s43545-022-00315-w
  • BartImmerzeel, B. et al. (2022). Appreciation of Nordic landscapes and how the bioeconomy might change that: Results from a discrete choice experiment. Land Use Policy 113, https://doi.org/10.1016/j.landusepol.2021.105909
